# VARIABLE RESOLUTION:
# IMPROVING SCENE VISUAL QUESTION ANSWERING WITH A LIMITED PIXEL BUDGET

## ABSTRACT

Artificial intelligence (AI) scene understanding systems can benefit from utilizing a large visual field of view (FOV). Some existing systems already employ multiple cameras to extend their FOV. However, increasing image size and quality present an overwhelming challenge to the acquisition and computing resources for such systems. An effective solution is to sub-sample the FOV, without impairing the model's performance on complex visual tasks, that require the integration of fine visual cues with contextual information. In this paper, we show that a variable sampling scheme, inspired by human vision, remarkably outperforms a uniform sampling scheme in the challenging task of scene visual question answering (VQA), under a limited samples budget (3% of the full resolution baseline). In particular, we demonstrate an accuracy gain of 3.7%, 2.0% and 0.9% on the GQA, VQAv2 and SEED-Bench datasets, respectively. The improvement is achieved without any image scanning and the variable resolution peaks at an arbitrarily chosen fixed image location. To understand the representational roots of this performance gain in complex vision-language models, our study also compared basic visual sub-tasks, in particular object classification and object detection. Comparing the variable and uniform models revealed differences in the representations, such as attention, learned by the different models, which yield a consistently improved performance of the variable resolution models. We show that the variable sampling scheme allows the models to benefit in low resolution areas, by propagating information from the finer resolution areas, and at the same time higher resolution areas benefit from contextual information at lower resolution in the periphery. The results show the potential of the biologically-inspired image representation to improve the design of visual acquisition and processing models in future AI-based systems.

## 1 INTRODUCTION

There is consistent evidence in nature that highly developed species utilize large field of views (FOV) to cope with fundamental visual tasks, including efficient detection of danger, food and social agents (Read, 2021; Martin, 2007). Similar to biological systems, an artificial intelligence (AI) system, designed to operate autonomously in natural environments, will require a visual system with a large FOV. Indeed, self-driving cars utilize multiple video cameras at various viewing directions, to gain a wide FOV of the surroundings (Nguyen et al., 2022). On the other hand, current deep learning models applied to various visual tasks at increasing complexity, from object detection to scene interpretation, utilize images at increasing size and quality, which present a great challenge to the acquisition and compute resources in AI-based systems (Pang et al., 2021; Said & Barr, 2021).

In nature, many species developed general-purpose visual systems that effectively cope with tasks requiring fine details at high resolution. For example, the human FOV spans over 120°, with peak resolution approaching 0.5 arcminutes (Marr et al., 1980). To cover such a large FOV at the finest resolution, an acquisition system would require more than 200 million samples. Anatomically, such a requirement is infeasible, requiring the diameter of the optic nerve (from the eye to the brain) to increase in diameter from 3-5mm to almost 10cm (Sylvester & Ari, 1961). For artificial vision systems, this requirement would involve the use of expensive digital apparatus and a heavy load on

Q: Where do these animals live?

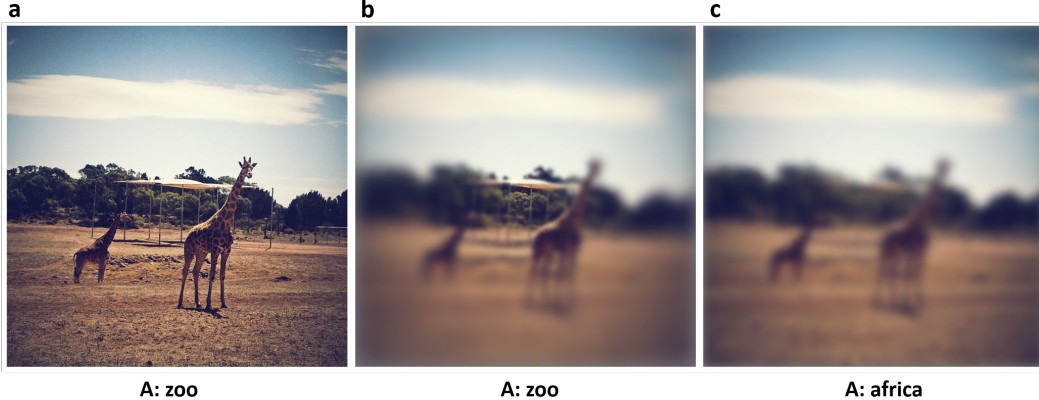

Figure 1: **Visual question answering example. (a)** baseline full resolution, **(b)** variable resolution, **(c)** uniform resolution. The model yields the correct answer when applied to the variable resolution image, with a mere 3% sample budget. While the uniform resolution is sufficient to recognize the giraffes (which normally live in Africa), only the variable resolution provides the fine details of the artificial shade, which is critical to answer the question correctly.

multi-sensor data fusion (Yeong et al., 2021). In addition, the visual processing with current visual models would require a scaling of the processing power by several orders of magnitude (Mnih et al., 2014).

A simple solution to these challenges is to sub-sample the FOV, *i.e.*, picking pixels at sparse locations such that the total count remains significantly low. Evolution provided an elegant and efficient solution in the form of a variable-resolution visual system, which acquires images at high resolution only in a small region at the center of the visual field (a.k.a. fovea centralis). In the rest of the visual field, resolution decreases as a function of eccentricity (distance from the center). Nevertheless, the brain provides mechanisms to extract useful information (e.g., context and scale) from the periphery at low resolutions that combines with the high-resolution foveal information, which together equally facilitate the visual perception (Trevarthen, 1968; Bar, 2003; Han et al., 2017; Oliva & Torralba, 2007).

In this work, we study the behavior of deep neural network models performing complex visual tasks when applied to variable resolution images, inspired by the human visual system, under an extremely limited pixel budget. In particular, we focus on the task of visual question answering (VQA). The VQA task is highly related to the task of complex visual scene understanding, for which we hypothesize that the contribution of a variable resolution system is highly significant (see example in Fig. 1). Understanding object relations and interactions in scenes, as well as object attributes, requires the integration of fine visual cues with contextual information, which are both available with the variable resolution scheme. To emphasize the general advantage of the variable sampling scheme, we focus on models' evaluation at a single fixation, without any scanning, where the variable resolution peaks at an arbitrarily chosen fixed image location, not aligned with the objects layout in the image.

The main contributions of this paper are as follows:

1. Large vision-language foundation models (VLMs) provide the state-of-the-art performance on the task of VQA. We apply three VLMs on three VQA datasets, comparing between the variable and uniform sampling schemes. In particular, we evaluate the models ViLT (Kim et al., 2021), MDETR (Kamath et al., 2021) and BLIP2 (Li et al., 2023b) on VQAv2 (Goyal et al., 2019), GQA (Hudson & Manning, 2019) and SEED-Bench (Li et al., 2023a) datasets, respectively. We show that the models improve significantly for most question types with the variable resolution images compared to the naive uniform resolution alternative. Furthermore, with a mere 3% pixel budget, models reach about 80% accuracy in comparison with the full resolution baseline.

**(a)**    **(b)**

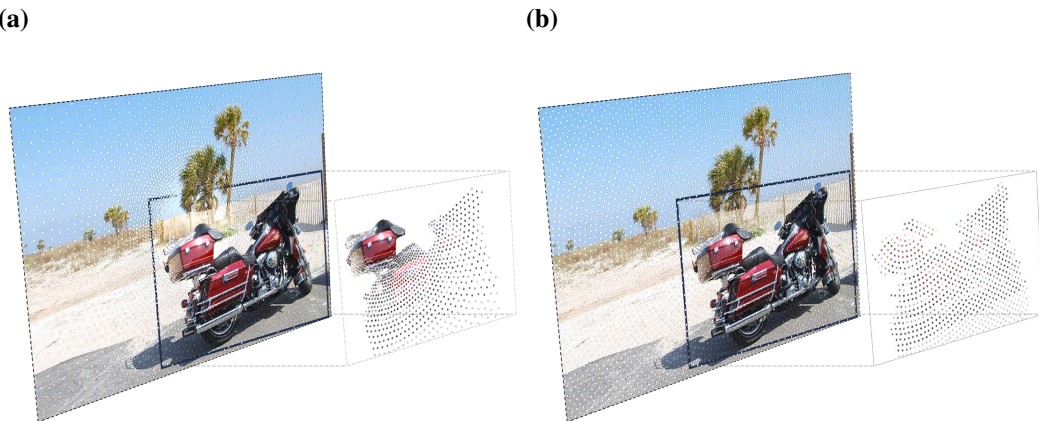

Figure 2: **Alternative sampling schemes. (a) Variable resolution** with peak sample density at the center of fixation (image center in this paper) and linearly decreasing number of samples with eccentricity. **(b) Uniform resolution** with a constant density of samples. Both sampling schemes distribute an equal number of samples over the entire FOV (we use log-polar coordinates). Interpolation is applied to get back an image at its original dimensions. In this work we address the question: which of the two alternative (motorbike) representations improves on complex visual tasks with existing DNN architectures, given that both alternatives consist of an equal number of samples?

2. In VQA many questions are around objects, their attributes and relations, hence requiring the capacities of sub-tasks such as object classification and detection. We train models for these tasks with each of the sampling schemes, revealing different representations for the variable and uniform schemes. Our experiments show improved performance of the variable over the uniform resolution models in both tasks. Moreover, the results indicate that the variable sampling scheme allows the models to benefit in low resolution areas, by disseminating information from the finer resolution areas, and in higher resolution areas from contextual information at low resolution in the periphery.

3. We examine aspects of interpretability in the models trained with mixed-resolution by comparing the internally learned features and showing the underlying filter specialization emerging from the training process. We demonstrate how these learned representations guide the attention of the models, resulting in improved performance for classification, detection and VQA.

## 2 EXPERIMENTAL SETUP

We consider three main sampling configurations of the input images given, and train a different model for each of the input sampling configurations. Importantly, we note that for the sub-sampling techniques (variable and uniform) we utilize a simple bilinear interpolation to form the final image while preserving spatial alignment. This allows us to extensively test existing vision systems made to work with Certasian coordinates without performing architectural changes.

**Baseline.** We refer to the given original images as "baseline" or "full resolution", utilizing 100% of the available pixel budget. The FOV spanned by the original image pixels is referred to as the "full FOV". We do not apply any pre-processing to the baseline images during inference or training. As such, in the context of our experiments, they represent the performance achievable by sampling the FOV at the highest available resolution. We extensively refer to this baseline for comparison purposes.

**Variable sampling.** The sampling approach follows Poggio et al. (2014) and Wilson & Bergen (1979), which modeled the human representation of visual information in the retina and the visual cortex. The *variable resolution* scheme, consists of sampling with a receptive field size, which continually increases with eccentricity (Fig. 2a). We apply a Log-Polar transformation to each image, where sample density remains constant $\forall \theta \in [0, 2\pi]$ and decreases linearly with $r$

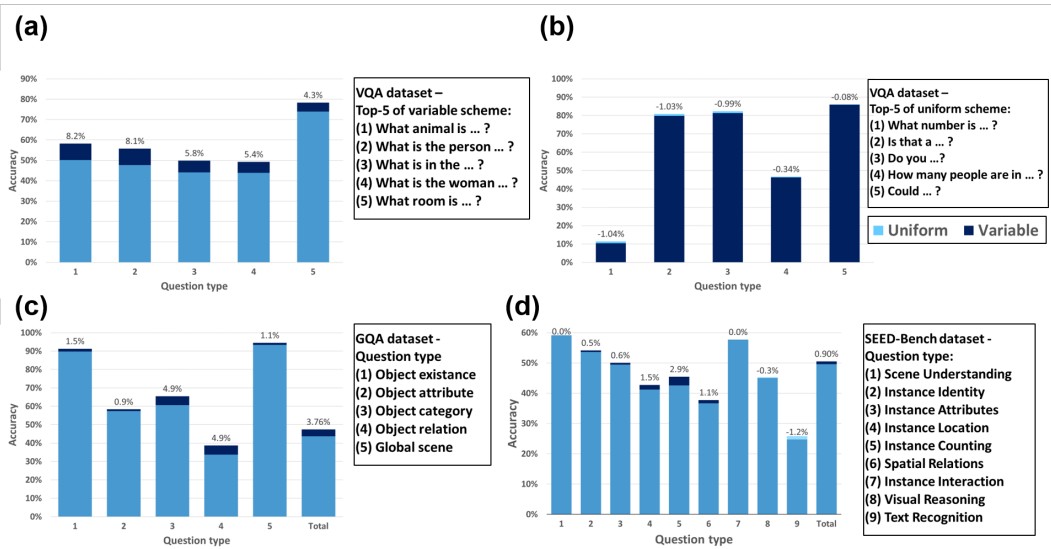

Figure 3: **VLMs' evaluation on visual question answering datasets.** (a,b) Accuracy on the VQAv2 dataset (ViLT): (a) The top-5 questions where the variable scheme outperforms the uniform, (b) where the uniform outperforms the variable. The Variable advantage is high and statistically significant, the uniform advantage is not, showing that the variable is consistently better. (c) Accuracy on the GQA dataset (MDETR). (d) Accuracy on the SEED-Bench dataset (BLIP2). In all models and almost all tasks the variable scheme performs significantly better. Numbers above the bars indicate the marginal accuracy difference

$$(r, \theta) = (log(\sqrt{(x - x_f)^2 + (y - y_f)^2}), arctan(\tfrac{y - y_f}{x - x_f}))$$

where $x_f, y_f$ are the coordinates of the fixation point (image center, in this paper). See Figure 2a. The sampling yields a budget of about 10K samples (pixels) within the full FOV. In the context of the COCO dataset (Lin et al., 2014), with typical image size of $480 \times 640$, this amounts to a pixel budget of about 3%. Note that throughout this paper, we arbitrarily picked the center of the image as the location of the highest sample density, regardless of the task or object location in the scene.

**Uniform sampling.** In this sampling approach the budget of samples is uniformly distributed across the entire FOV. We employ a concentric grid to comply with the variable sampling. This *uniform resolution* schema is akin to simply down-scaling the FOV. See Figure 2b.

**Training paradigm.** We trained all object detection and classification models (Mask-RCNN, DETR, ResNet18/50/101, SqueezeNet (He et al., 2017; Carion et al., 2020; He et al., 2016; Iandola et al., 2016)) from scratch using their original hyperparameters and training methods. Each model was trained on one of three versions of its dataset, matching the sampling configuration used in testing. As such, we have three models for every experiment.

**Interpretability analysis.** The interpretability analysis for object detection was conducted using the DETR model. This model is combined with a BERT-based LLM in the MDETR VLM in our VQA experiment on the GQA dataset.

## 3    VISUAL QUESTION ANSWERING (VQA)

Visual question answering is a complex yet one of the most fundamental visual tasks for an intelligent agent. This task requires the perception of subtle cues related to object relations, interactions and causality in a scene. The VQA task combines multiple mechanisms to answer questions about specific parts of the image, the general layout of the scene or both. In our experiments, we first applied a pretrained ViLT model, on the VQAv2 dataset consisting of 65 question types. We evaluated the model on three sampling schemes: baseline-full resolution, variable and uniform. The results clearly indicate a significant advantage for the variable sampling scheme, with an overall ac-

curacy gain of 1.96% on the validation set (1.44% on the test set) compared to the uniform scheme ($M_{var} = 64.9 \pm 19.8\%$ ; $M_{uni} = 62.9 \pm 19.9\%$ ; $t[64] = 9.16$ ; $p < 1 \times 10^{-6}$, on the validation set). The mean accuracy for the baseline with full resolution images on this dataset is 81.1%.

Next, we applied the MDETR model on the GQA dataset, which demonstrated similar trends. The model evaluated on the variable sampling scheme achieved a total accuracy of 47.4% compared to 43.7% for the uniform scheme ($t[4] = 2.32$ ; $p = 0.04$). The GQA 'testdev' set consists of 12,578 questions and 398 images. In a third evaluation we tested the BLIP2 model on the new SEED-Bench dataset, where the variable scheme outperforms the uniform by 0.9% (50.5% vs. 49.6% total accuracy; $t[14232] = -6.00$ ; $p < 1 \times 10^{-6}$). Due to computational resource limitations we fine-tuned only the MDETR model among the VLMs. The training indeed yielded an improved performance for both sampling schemes (see Supplementary). The fact that the variable scheme consistently outperformed the uniform scheme already in the fine-tuned models, mitigates the need to re-train also the BLIP2 and ViLT models.

As shown in Figure 3, most question types benefit from the variable sampling scheme, which provides finer details in the center of the image, over the uniform scheme. The advantage is even more remarkable when considering the fact that in our experiment we arbitrarily selected the center of the image as the highest resolution area in the variable resolution images, regardless the objects' layout in the scene, while the cues for answering the questions can be located anywhere in the scenes. Interestingly, we note that the Top-5 marginal difference question types on VQAv2 have a very different distribution for both the uniform and the variable schemes. The Variable resolution scheme yields marginal gains of up to 8.2% (see Fig. 3a). On the contrary, the uniform sampling scheme prevails with *at most* 1.0% and in most cases orders of magnitude less (see Fig. 3b). The same pattern is evident in both the GQA and the SEED-Bench datasets (Fig. 3c,d).

## 4 OBJECT DETECTION

As a gateway to explore the results achieved on the complex VQA task, we evaluated the behavior of several models on the underlying task of object detection, which is essential for VQA. For this task, we utilized the DETR (also used for VQA) and Mask-RCNN architectures with segmentation heads.

**Image classification on Imagenet.** Both models used a ResNet101 backbone, pretrained on ImageNet-1K, following the variable and uniform sampling schemes. Already at this stage, the variable resolution model improves over the uniform by more than 10% (Supplementary, Table S6).

Table 1: **Object detection evaluation.** Bounding box performance metrics on COCO validation set.

| Model | Sampling | AP | AR | $AP_S$ | $AP_M$ | $AP_L$ |
|---|---|---|---|---|---|---|
| DETR | Baseline | 42.6 | 58.3 | 21.7 | 46.9 | 60.7 |
| DETR | Uniform | 21.8 | 36.1 | 4.0 | 19.4 | **40.3** |
| DETR | **Variable** | **22.6** | **36.6** | **6.2** | **20.7** | 40.2 |
| MaskRCNN | Baseline | 40.1 | 52.7 | 22.7 | 43.0 | 52.7 |
| MaskRCNN | Uniform | 18.1 | 30.5 | 3.9 | 17.3 | 32.0 |
| MaskRCNN | **Variable** | **19.3** | **31.7** | **6.0** | **18.3** | **32.0** |

**COCO Dataset.** The Mask-RCNN and DETR models were then trained on COCO and evaluated on the COCO-2017-val. The results show that the variable resolution models consistently outperform the uniform resolution models (Table 1). The gain for medium size objects is in particular interesting, since only a small part of the objects can get covered by the high resolution area, while most of objects must fall in the peripheral lower resolution areas. This suggests the dissemination of the high-resolution information within the model.

**Variable resolution improves performance in low-resolution areas.** The two datasets under review, ImageNet-1K and COCO, mostly concentrate on objects positioned at the center of the image, directly aligning with our high-resolution area.

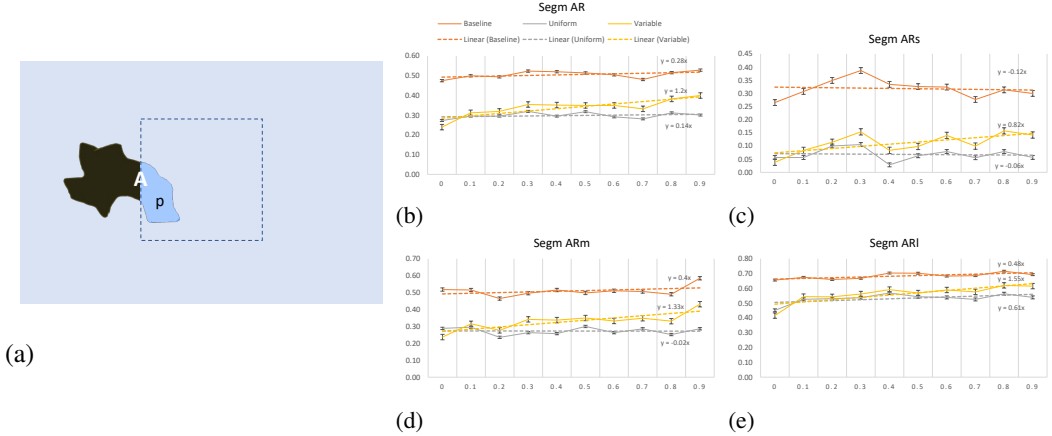

Figure 4: **Bin experiment.** (a) The $\frac{P}{A}$ HRR allows for measuring the degree to which a ground-truth object is contained in the high-resolution middle area. (**b-e**) Evaluation variable vs uniform sampling performance w.r.t. degree of inclusion (HRA = $200 \times 200$). We performed 5 randomized trials for each $G_{i-i+0.1}$, hence the error bars. We applied a *paired t-test* to the variable and uniform model accuracies and were able to reject the null hypothesis ($H_0$ : no significant difference between the models) for $\alpha = 0.05$ every time.

Consequently, it is not surprising that the variable-resolution model gains an edge in performance due to high-resolution input focused on critical image areas. At this juncture, the reader might be tempted to accredit the benefits of variable resolution to merely coincide with the pre-existing photographer bias in the information-dense region of these datasets.

To investigate these concerns in-depth, we devised experiments aimed at (1) tracking the models' performance in relation to how much of the high information density area objects occupy, and (2) examining the models' performance on objects enveloped by an identical sample count, varying only in its distribution pattern.

Table 2: **Sample-equalised evaluation.** Segmentation accuracy measured only for objects covering an equal number of samples with both variable and uniform sampling schemes.

| Model | Sampling | AR | $AR_S$ | $AR_M$ | $AR_L$ |
|---|---|---|---|---|---|
| DETR-R101 | Baseline | 54.8 | 15.3 | 51.9 | 72.0 |
| DETR-R101 | Uniform | 37.1 | 1.2 | 28.0 | 51.6 |
| DETR-R101 | **Variable** | **38.5** | **2.1** | **31.0** | **54.7** |
| Mask RCNN-R101 | Baseline | 52.5 | 25.6 | 49.7 | 64.2 |
| Mask RCNN-R101 | Uniform | 34.7 | 1.6 | 27.7 | 48.2 |
| Mask RCNN-R101 | **Variable** | **36.9** | **3.4** | **31.7** | **48.6** |

**Creating annotation bins (1).** Take the set of all images in the COCO validation set $V = \{I_1, I_2...I_{5,000}\}$, $I_i \in \mathbb{Z}^{W_i \times H_i \times 3}$. Define a square, $D \times D$ in dimension ($D \in \mathbb{Z}$) s.t. $D < W_i$, $D < H_i \; \forall i \in \{1...5,000\}$. E.g. $D$ could take a value of 200, since even the smallest images in COCO have a larger spatial size. We now center such a square of fixed size on every validation image and call the parts of the image contained inside of it the high-resolution area (*HRA*), since they contain our highest information density.

For each ground truth annotation, we then simply calculate $\frac{P}{A}$ as the fraction of its pixel mask area inside the *HRA* (Figure 4a). We call this metric the high-resolution inclusion degree (*or inclusion degree* for short). It expresses the degree of inclusion of that annotation in an *HRA* of a given size, providing a way to measure the advantage a model utilizing variable sampling would have in

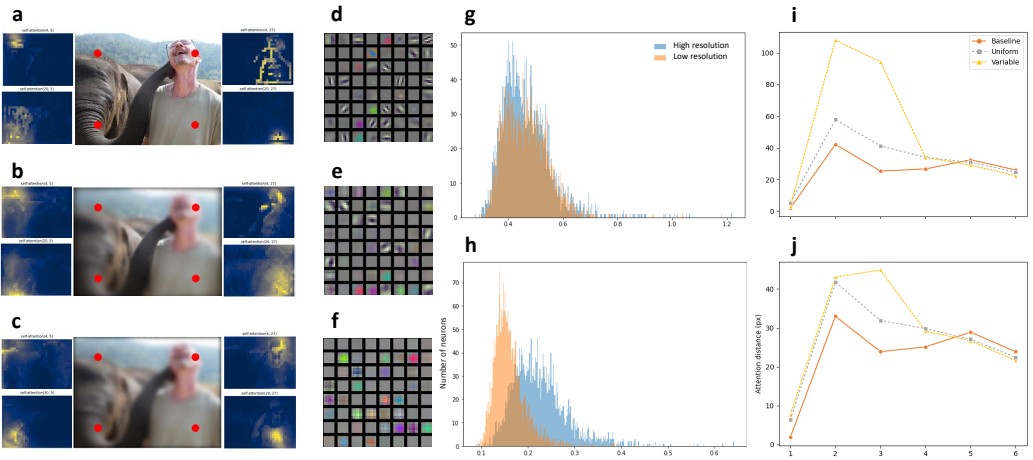

Figure 5: **Interpretability (a, b, c)** Example visualizations of the highest self-attention layer of the DETR transformer model for several tokens located at the periphery. The red dots indicate the tokens' spatial locations. The corresponding attention maps are presented to the left or right of the image (closest to the associated red dot): (a) baseline-full, (b) variable and (c) uniform sampling schemes. **(d, e, f)** Kernel filters of the first convolutional layer for full, variable and uniform resolutions, respectively. **(g, h)** Neuronal activation maps for intermediate layers on high and low resolution areas, showing differences in the feature maps. **(i, j)** Attention distance charts for query tokens centered at the central high resolution area (i) and at the peripheral lower-resolution areas (j), showing large distances in attention spread, see text.

detecting/segmenting that object. Notice that the value of the inclusion degree for every annotation depends on the parameter $D$. For example, the object in Figure 3A may have an inclusion degree of 0.5 for a *HRA* of $200 \times 200$, but an inclusion degree of 0.3 for a $150 \times 150$ *HRA*. For the purposes of our experiments, we tested *HRA* sizes ranging from $100 \times 100$ to $250 \times 250$, in size increments of 10 pixels per dimension (16 *HRA* sizes in total) and achieved similar results on all of them.

We now assign an inclusion degree bin to every annotation in the validation set based on its inclusion degree value. The set of ground truth annotations $G_{0.0-0.1}$ contains only annotations with an inclusion degree *between 0.0 and 0.1*, including the most peripheral objects only. The set of annotations $G_{0.1-0.2}$ includes only inclusion degrees between 0.1 and 0.2 and so on. We define ten (10) of those *bins*: $G_{0.0-0.1}$, $G_{0.1-0.2}$,... $G_{0.9-1.0}$. Their union gives back the original validation set. Figure 4(b-e) shows the performance of the models on the bins. The pattern from the VQA is also reflected in those results; the variable sampling benefits a lot when objects are included in the high-information density area, whereas the advantage of the uniform sampling approach is very limited, even in very peripheral bins, where the uniform sampling has an advantage from the sample-count standpoint.

**Counting samples (2).** As an additional control, we constructed a sample-equalized validation set by counting how many samples fall within the annotation mask of each object (see Fig. 2). We report results on the sample-equalized validation set in Table 2, where we see a similar performance gap. This, in tandem with our bin-evaluation experiments, further emphasizes the advantage that having variable resolution gives.

## 5    ON INTERPRETABILITY

At this point we turn to investigate the models internal representations to discover what drives the behavior observed in the former experiments. In particular, we would like to explore the differences in the internal learned representations between the two sampling schemes, including filter kernels and neuronal activation. In addition, we compare the effect of these learned representations on the self attention of transformer models trained to detect and segment objects.

**(1) Single model generalizes to detect multi-resolution-spanning objects by allowing resolution-specialization in neurons.** The convolutional kernels in CNN-based models are commonly applied

uniformly across the entire image. It is therefore not clear if and how CNNs adapt when presented with multiple resolutions within the same object instance. We have demonstrated superior performance of the variable sampling approach, and looking-under-the-hood we now pose the question: what internal adaptations occur in the model to facilitate beneficial information-flow from low to high-resolution parts of a single object. We hypothesized that the variable resolution trained models learned a mixed resolution representation, where some neurons specialize on low-resolutions and other neurons on high-resolutions. Whether this occurs is not a trivial question, since it is entirely possible that the neurons of the network have learned the average resolution in our training set only: producing high dot-products (activations) for mid-resolution occupying object segments and low everywhere else.

We now refer the reader to Figure 5g,h. The histograms show the maximal neuronal activation values in several filters from the deep layers of the ResNet101 architecture, backbone for our main detection models (MaskRCNN and DETR). The network was trained under the variable resolution sampling scheme and applied on an equal number of crops from peripheral (primarily low-resolution) and central (primarily high-resolution) locations, both from the variable resolution COCO validation set. Note importantly that the crops inferred to the network had an identical distribution of object types, sizes, and extraction locations. The only difference between the crops was the subset of our variable resolution continuum they contained. We refer the reader to the Supplementary for more details.

We observe that while some filters seem universal (Figure 5g) w.r.t. the resolution objects span, other filters (Figure 5h) exhibit *selectivity*, firing actively under certain resolution conditions only. This indeed suggests that the network dedicates groups of neurons (filters) to certain resolutions only, and others more universally.

**(2) Variable resolution induces attention spread in transformer architectures.** As a qualitative result, we now refer the reader to (Figure 5a-c). Here we provide some intuition behind the internally developed representations of the transformer module in DETR, which also utilizes the ResNet101 architecture we discussed above and is used in VQA.

We observe that for the uniform resolution model, the self-attention appears to act more locally, whereas the variable resolution appears to induce an *object-centric* wide spread of the attention, feeding more on the contextual cues provided by the diverse context in the image (texture from the middle, scale and location from the periphery).

As a quantitative metric, we followed Park et al. (2023) and computed the average attention distance in pixels for each transformer encoder layer in the uniform, variable and baseline DETR models. We measured this metric in central and peripheral image regions. We define the peripheral attention distance as the attention distance of query tokens in the outer-most 30%-35% of the image. This is roughly the location of the query (red) points in Fig 5a-c. The central attention distance is defined as the attention distance of query tokens *only* located in the middle 10% of the image, at the high-resolution area. Figure 5i,j show that the peripheral and central attention distances for the variable model extend further, whereas the uniform model is more local. This corresponds to the qualitative representations above, showing a reciprocal informational exchange from center to periphery and vice-versa.

## 6 Related Work

Prior computational work explored different aspects of non-uniform sampling in the visual FOV, including foveal schemes, where samples are distributed densely around a fixation point in the FOV and more sparsely in the periphery (Akbas & Eckstein, 2017; Lukanov et al., 2021; Paula & Moreno, 2023). Early studies, such as by Freeman & Simoncelli (2011), developed models of the human visual system to evaluate the capabilities and limitations of human peripheral vision, but did not address the implications of these models on artificial systems. Based on this study, Deza & Konkle (2021) studied the impact of foveated texture-based input representations in artificial vision systems on the task of scene classification. They compared between a few matched-resource models: Foveation-Blur, Uniform-Blur (similar to the schemes in Section 2) and Foveation-Texture. They showed that peripheral texture encoding leads to representations with greater generalization, sensitivity to high-spatial frequency and robustness to occlusion. Wang & Cottrell (2017) explored a

neurocomputational modeling of central and peripheral vision for scene recognition. They suggested that the advantage of peripheral over central vision is due to intrinsic usefulness of features carried by peripheral vision, generating a greater spreading transform in the internal representational space. They predicted that the two pathways correlate with their neural substrates, LOC and PPA in the brain. However, scene classification may provide only limited insight, as it can be often performed well at extremely poor resolutions (Torralba et al., 2008).

Pramod et al. (2022) suggested that blurry peripheral vision may have evolved to optimize object recognition. They experimented with DNNs applied to foveated images around objects of interest, showing that the networks performance is peaked at the human blur decay setting, also benefiting from reduced false detections in the blurry periphery. Other studies investigated the effects of *cortical magnification*, a brain mechanism that allocates more processing units to the densely sampled area of the foveal image. Jaramillo-Avila & Anderson (2019) used such foveated videos to fit models into embedded systems. They achieved a 4× speed-up in frame rate, but show only a small decrease in recall within the restricted foveal region. Similarly, Kundu et al. (2020) developed a real-time system based on 'magnified' foveated images for urban scene parsing (semantic segmentation) targeting self-driving cars.

A key limitation of these studies is the absence of a robust comparison with large vision models used on large-scale datasets such as COCO. Furthermore these studies lack a comparison with prevalent architectures such as the transformer. Lastly, none of the studies provided a systematic analysis on models performance for objects mostly contained in the periphery rather than in the higher resolution area.

## 7 CONCLUSIONS

We address the problem of current artificial vision systems in covering a large FOV, while enabling the acquisition of fine details in high resolution to perform complex visual tasks. Inspired by the human visual system, we employ and evaluate a variable resolution sampling scheme, under a limited budget of samples, with a high resolution area at the center of the FOV and linearly decreasing resolutions in the periphery.

When applied to the complex VQA task, the variable sampling scheme consistently outperforms the uniform sampling scheme across most question types, as demonstrated in Section 3 for the ViLT, MDETR and BLIP2 VLMs on VQAv2, GQA and SEED-Bench datasets, respectively. This is an outstanding finding, mainly from two perspectives. First by considering the fact that we arbitrarily choose the highest resolution area location in the center of the image, while the cues required to answer the questions can be anywhere in the scene. Second, the improvement is achieved with a single fixation, without any scanning across the FOV. In early studies, Torralba et al. (2008) showed that humans and machines can perform well on the task of scene classification only with a uniformly sub-sampled *gist* of an image. Our results indicate that a variable resolution scheme, is a better alternative than the uniform sampling scheme (e.g., the question "what room is?" in VQAv2 yields an improvement of 4.3%, the "object relation" question type in GQA and the "instance counting" question type in SEED-Bench, both gains 2.9%; see Fig. 3).

This gain in performance at a single arbitrary fixation, suggests the dissemination of high resolution information from the center of the FOV to the periphery and of low resolution contextual information from the periphery to the center (see section 5 and Fig. 5) . Our experiments on image classification and object detection show the source of this dissemination, in the internally learned representations of the models. Kernel filters at the lower convolutional layers of the backbone become specialized for high and low frequencies in the variable resolution models (Fig. 5d-f). Additionally, neurons in intermediate layers become specialized for the mixed frequencies in the input (Fig. 5g,h). Remarkably, high-resolution selective kernels and neurons are learned even despite the small high-resolution area, compared to the large peripheral area at lower resolutions. Lastly, we demonstrate how these internal representations affect the self-attention of transformer models, clearly showing an improved spread of attention for objects covering high and low resolution areas in the FOV (Fig. 5i,j).

Overall, the results show the potential of the biologically-inspired image representation to improve the design of visual acquisition and processing models in future AI-based systems.

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

# SUPPLEMENTARY MATERIALS

## A    DETAILED RESULTS OF THE VQA EXPERIMENTS

Table 3: **MDETR model evaluation on GQA testdev dataset.**

| Model | Question type | Acc. Baseline (full) | Acc. Variable | Acc. Uniform |
|---|---|---|---|---|
| Pretrained | (1) Object existence | 95.6% | **91.3%** | 89.8% |
| | (2) Object attribute | 71.2% | **58.3%** | 57.4% |
| | (3) Object category | 76.0% | **65.5%** | 60.6% |
| | (4) Object relation | 53.1% | **38.6%** | 33.7% |
| | (5) Global scene | 95.8% | **94.5%** | 93.3% |
| | Total | 61.7% | **47.4%** | 43.7% |
| Fine-tuned | (1) Object existence | 95.6% | **93.8%** | 93.4% |
| | (2) Object attribute | 71.2% | **62.6%** | 62.6% |
| | (3) Object category | 76.0% | **71.1%** | 70.3% |
| | (4) Object relation | 53.1% | **46.5%** | 44.9% |
| | (5) Global scene | 95.8% | 95.2% | **95.7%** |
| | Total | 61.7% | **54.3%** | 53.3% |

Table 4: **BLIP2 model evaluation on SEED-Bench dataset.**

| Model | Question type | Acc. Baseline (full) | Acc. Variable | Acc. Uniform |
|---|---|---|---|---|
| Pretrained | (1) Scene Understanding | 59.3% | 59.1% | **59.2%** |
| | (2) Instance Identity | 53.7% | **54.2%** | 53.6% |
| | (3) Instance Attributes | 49.4% | **50.0%** | 49.5% |
| | (4) Instance Location | 42.8% | **42.7%** | 41.2% |
| | (5) Instance Counting | 43.0% | **45.4%** | 42.5% |
| | (6) Spatial Relations | 36.5% | **37.8%** | 36.7% |
| | (7) Instance Interaction | 55.7% | **57.7%** | 57.7% |
| | (8) Visual Reasoning | 45.0% | 45.0% | **45.3%** |
| | (9) Text Recognition | 27.1% | 24.7% | **25.9%** |
| | Total | 49.8% | **50.5%** | 49.6% |

Due to the very large dimensions of the SEED-Bench dataset images, we used in the experiment reported in Table 4, a different sampling budget for both the variable and uniform schemes, in particular using 15% instead of 3% of the full-resolution samples budget, that was used in all other experiments in this study.

We note that for some question types of the SEED-Bench dataset, the variable resolution scheme outperforms the baseline full-resolution (e.g., "(5) Instance Counting"). This improvement is not intuitive, but may be possible under some conditions. we think that these results require further examination to better understand the conditions under which this advantage of the variable over the full resolution scheme can be better understood.

Table 5: **ViLT model on VQAv2**$_{val}$

| Question Type | Baseline Acc. | Variable Acc. | Uniform Acc. | $\triangle$ |
|---|---|---|---|---|
| What animal is | 96.86% | 58.28% | 50.13% | 8.2 % |
| What is the person | 57.93% | 55.81% | 47.72% | 8.1 % |
| What is in the | 97.34% | 49.86% | 44.08% | 5.8 % |
| What is the woman | 92.26% | 49.26% | 43.85% | 5.4 % |
| What room is | 94.10% | 78.33% | 74.00% | 4.3 % |
| What sport is | 94.97% | 80.27% | 76.34% | 3.9 % |
| What is this | 18.62% | 56.32% | 52.80% | 3.5 % |
| Who is | 44.28% | 47.06% | 43.56% | 3.5 % |
| What is | 95.59% | 45.67% | 42.36% | 3.3 % |
| What is the man | 93.86% | 53.79% | 50.72% | 3.1 % |
| Do | 81.99% | 80.95% | 77.94% | 3.0 % |
| Is this person | 71.30% | 85.75% | 82.93% | 2.8 % |
| What are the | 95.06% | 49.54% | 46.75% | 2.8 % |
| What type of | 96.80% | 55.70% | 52.94% | 2.8 % |
| What color is | 91.81% | 78.05% | 75.33% | 2.7 % |
| What color are the | 95.61% | 71.93% | 69.25% | 2.7 % |
| What kind of | 67.16% | 55.99% | 53.33% | 2.7 % |
| Which | 92.90% | 54.17% | 51.67% | 2.5 % |
| What are | 95.58% | 52.77% | 50.28% | 2.5 % |
| Are | 58.02% | 79.86% | 77.40% | 2.5 % |
| How many | 88.85% | 48.79% | 46.44% | 2.4 % |
| Is this an | 51.02% | 86.01% | 83.76% | 2.3 % |
| What is the | 94.08% | 50.50% | 48.25% | 2.3 % |
| Is he | 77.66% | 83.62% | 81.42% | 2.2 % |
| What is the color of the | 79.40% | 79.52% | 77.40% | 2.1 % |
| Is the man | 64.60% | 82.07% | 79.97% | 2.1 % |
| Is the person | 60.59% | 81.32% | 79.28% | 2.0 % |
| What color | 93.01% | 72.72% | 70.70% | 2.0 % |
| What color is the | 71.91% | 75.69% | 73.70% | 2.0 % |
| What | 71.48% | 49.99% | 48.02% | 2.0 % |
| Are these | 92.47% | 84.82% | 82.87% | 1.9 % |
| Does this | 95.65% | 83.95% | 82.12% | 1.8 % |
| What does the | 76.51% | 30.32% | 28.57% | 1.8 % |
| What is the name | 96.96% | 21.67% | 19.94% | 1.7 % |
| Are the | 93.31% | 82.66% | 80.98% | 1.7 % |
| Is the | 94.08% | 83.31% | 81.66% | 1.7 % |
| None of the above | 66.71% | 65.69% | 64.05% | 1.6 % |
| Are there any | 95.03% | 80.80% | 79.19% | 1.6 % |
| Can you | 79.90% | 79.39% | 77.87% | 1.5 % |
| What is on the | 72.82% | 45.09% | 43.61% | 1.5 % |
| Are they | 93.73% | 82.37% | 80.90% | 1.5 % |
| Is there | 96.30% | 81.09% | 79.66% | 1.4 % |
| Is | 42.30% | 81.24% | 79.88% | 1.4 % |
| Does the | 73.88% | 80.55% | 79.21% | 1.3 % |
| Why is the | 88.96% | 33.89% | 32.61% | 1.3 % |
| Are there | 42.92% | 80.30% | 79.07% | 1.2 % |
| How | 82.95% | 42.38% | 41.23% | 1.2 % |
| Is this a | 82.14% | 85.28% | 84.20% | 1.1 % |
| Is the woman | 74.01% | 79.51% | 78.47% | 1.0 % |
| What brand | 95.82% | 48.33% | 47.34% | 1.0 % |
| Has | 44.81% | 83.11% | 82.22% | 0.9 % |
| Is this | 92.62% | 84.77% | 83.88% | 0.9 % |
| What time | 73.24% | 29.22% | 28.63% | 0.6 % |
| Why | 96.36% | 32.77% | 32.23% | 0.5 % |
| How many people are | 97.59% | 48.65% | 48.15% | 0.5 % |
| Where are the | 78.92% | 42.40% | 42.03% | 0.4 % |
| Where is the | 96.51% | 38.69% | 38.35% | 0.3 % |
| Is there a | 76.19% | 76.46% | 76.14% | 0.3 % |
| Is it | 97.60% | 88.41% | 88.27% | 0.1 % |
| Was | 96.98% | 86.05% | 85.99% | 0.1 % |
| Could | 29.58% | 85.92% | 86.00% | −0.1 % |
| How many people are in | 96.46% | 46.33% | 46.67% | −0.3 % |
| Do you | 91.19% | 81.33% | 82.32% | −1.0 % |
| Is that a | 88.89% | 79.80% | 80.83% | −1.0 % |
| What number is | 96.46% | 10.37% | 11.41% | −1.0 % |

## B    DETAILED CLASSIFICATION RESULTS

We begin this section by reporting our ImageNet-1000 classification results in greater detail. We tested several classification models under the variable and uniform sampling schemes. Table 6 shows our results.

Notably, we see that for the SqueezeNet CNN-based model, with just 1.2M parameters, the gap between the full resolution baseline and the variable resolution model performance is *smaller* than in the larger models (ResNet-s).

We hypothesize that sparser input (variable and uniform) benefits small architectures by internally requiring fewer neuronal resources in order to learn, being in a sense more compact for the neu-

Table 6: **Imagenet classification results.**

| Models | #params | Pixel budget | Sampling | Top-1 Acc. | Top-5 Acc. |
|---|---|---|---|---|---|
| ResNet-101 | 44M | 100% | Full (baseline) | 75.5 % | 92.6 % |
| ResNet-101 | 44M | 3% | Uniform | 53.4 % | 76.6 % |
| ResNet-101 | 44M | 3% | **Variable** | **66.3%** | **86.5%** |
| ResNet-18 | 11M | 100% | Full (baseline) | 68.8 % | 88.6 % |
| ResNet-18 | 11M | 3% | Uniform | 45.4 % | 69.4 % |
| ResNet-18 | 11M | 3% | **Variable** | **58.2%** | **80.6%** |
| SqueezeNet | 1.2M | 100% | Full (baseline) | 42.2 % | 65.7 % |
| SqueezeNet | 1.2M | 3% | Uniform | 27.6 % | 49.0 % |
| SqueezeNet | 1.2M | 3% | **Variable** | **35.9%** | **59.0%** |

ral processing system. We further demonstrate this neural representation compactness by showing that a single model trained on very sparse sampling input has significant excess neuronal resources (following section).

## C    DETECTION AND INTERPRETABILITY EXAMPLES

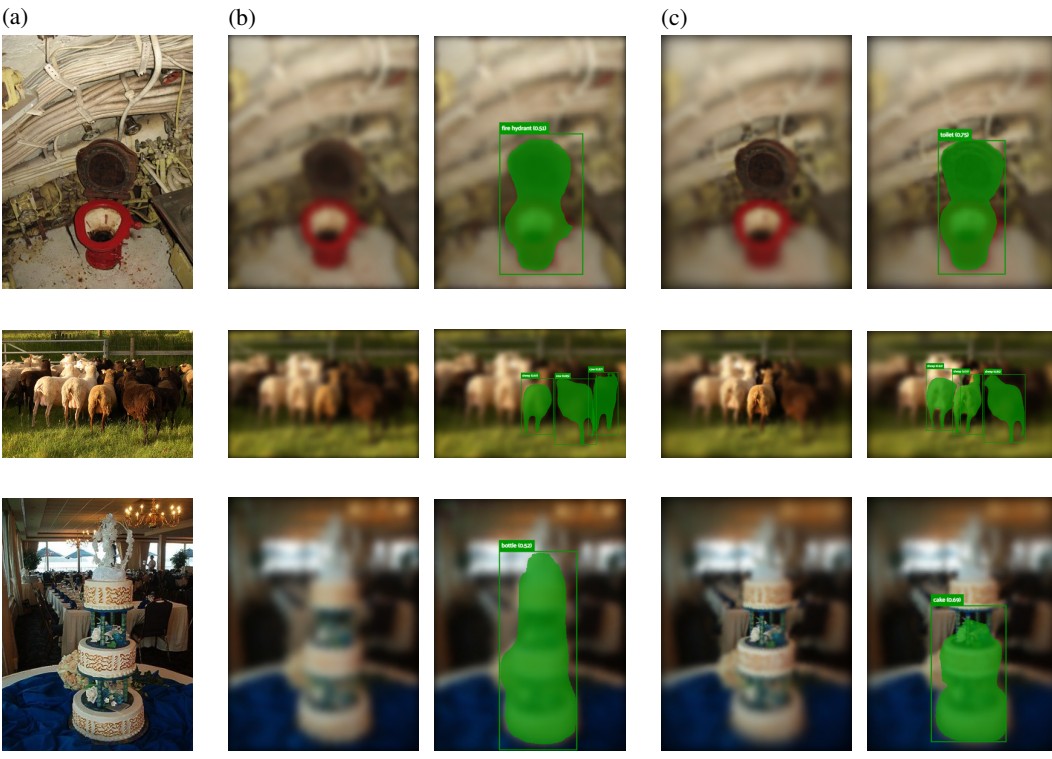

(a) (b) (c)

Figure S.1: **Detection examples of the DETR model** We see in several instances that the variable resolution model benefits from the texture of an object, critical for determining its class correctly. (a) Full-resolution image. (b) Uniform sampling scheme. (c) Variable sampling scheme.

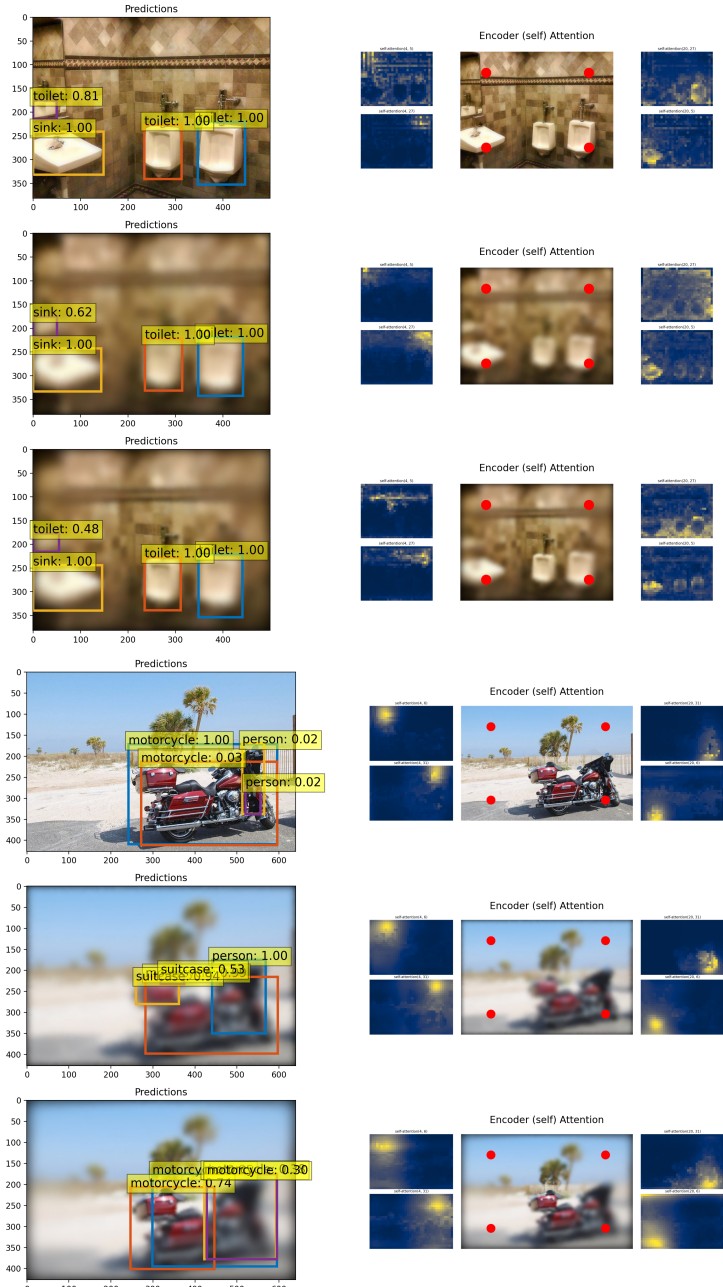

Figure S.2: **Additional interpretability and detection examples in DETR model.** Baseline full-resolution, uniform resolution and variable resolution.

# D  ABLATION EXPERIMENTS

## D.1  MULTI-MIXED MODEL

So far in the paper, we demonstrated a superior performance of variable sampling based models. We explored the internally learned features and showed some resolution *selectivity* within the nerons of the networks. We tested several additional model types, again trained and tested on variable resolution input, but with a discrete, rather than continuous, variability. Figure S.3a-S.3c shows the three different input types, all from the same initial original image in the COCO dataset. The Channel 1 input represents a $60 \times 60$ crop taken from the middle of the original image. The Channel

2 input represents a larger ($300 \times 300$) crop taken from the middle, with some uniform sampling applied. Channel 3 represents the original image, but with very sparse uniform sampling applied (sparser than the sampling we have been referring to as *uniform* until now). We have designed the filters applied to channels 2 and 3 such that the *total* number of samples in the three channels combined is equal to 3% of the total image samples. This allows us to explore the transfer learning effects on models trained with a similar pixel budget, as well as the utilization of neuronal resources occurring.

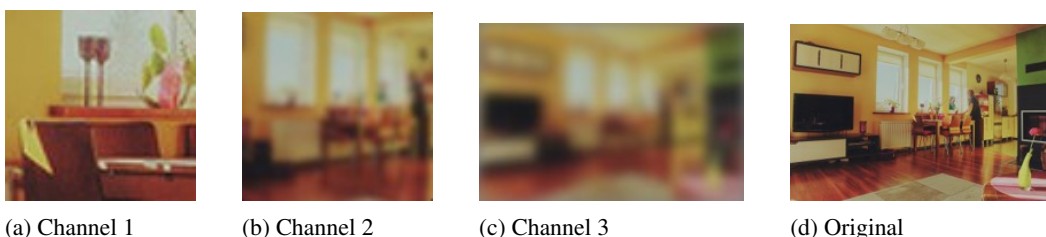

(a) Channel 1          (b) Channel 2          (c) Channel 3          (d) Original

Figure S.3: Multi-mixed channels input image examples

We trained 4 Mask RCNN-R101 models in total on the task of object detection (COCO); one model for each channel input and an additional model (called *mixed*) trained on all channels together. Each channel type has its own COCO version, containing only its appropriately sampled crops/images. The additional model, therefore, was trained on 3 combined COCO datasets with triple the number of images in the original training set.

This experiment allowed us to explore the transfer learning capabilities occurring in models trained on input with a variety of resolutions. Table 7 displays the results.

Table 7: **Object detection evaluation.** Bounding box performance metrics on COCO validation set.

| Model trained on | Tested on | AP | AR | $AP_S$ | $AP_M$ | $AP_L$ |
|---|---|---|---|---|---|---|
| $CH_1$ | $CH_1$ | 0.128 | 0.339 | 0.064 | 0.176 | - |
| $CH_2$ | $CH_2$ | **0.177** | **0.326** | **0.055** | **0.190** | **0.295** |
| $CH_3$ | $CH_3$ | 0.106 | 0.208 | 0.013 | 0.086 | 0.209 |
| $CH_{1+2+3}$ | $CH_1$ | **0.166** | **0.375** | **0.102** | **0.217** | - |
| $CH_{1+2+3}$ | $CH_2$ | 0.153 | 0.304 | 0.05 | 0.168 | 0.264 |
| $CH_{1+2+3}$ | $CH_3$ | **0.121** | **0.228** | **0.015** | **0.099** | **0.248** |

Surprisingly, we find that the mixed model outperforms the single-channel-trained models on their individual channels. This suggests two things: *(1)* the variable resolution approach induces beneficial transfer learning during training; *(2)* the networks trained on sparsely-sampled images have *excess, underutilized,* resources.

### D.2 NEURONAL SPECIALIZATION EXPERIMENT

We hypothesized that a network conditioned with variable resolution input will keep some separation between the neurons it dedicates to specific resolutions, even for a single object representation. To test our hypothesis, we fed the variable-resolution ResNet101 (backbone of our MaskRCNN and DETR models) images of Type 0 and Type 1 resolution (Figure S.4b-S.4c. Both of those resolutions are from the spectrum contained in the original training images, but the latter (Type 0 resolution) has its fixation shifted away from the center. As such, both images represent the same central crop, but in low variable resolution (Type 0), and high variable resolution (Type 1). We inferred 2,500 Type 0 and 2,500 Type 1 images (splitting the validation set in two) to MaskR-CNN. We extracted the feature map tensors produced by a deep layer in the ResNet101 backbone ($backbone.body.layer3.block22.batchnorm3$), which resulted in two sets of 2,500 tensors, with size varying from $1024 \times 50 \times 50$ to $1024 \times 70 \times 70$, depending on the size of the inferred image. For each 2D sub-matrix in the tensor (1,024 sub-matrices in total), we took the *average, median and*

*maximum* of the neronal activations. Those metrics serve as *descriptive statistics* for our following experimentation and reduce our datasets to *two sets* of $2,500$ tensors, each of which has dimension $1024 \times 3 \times 1$.

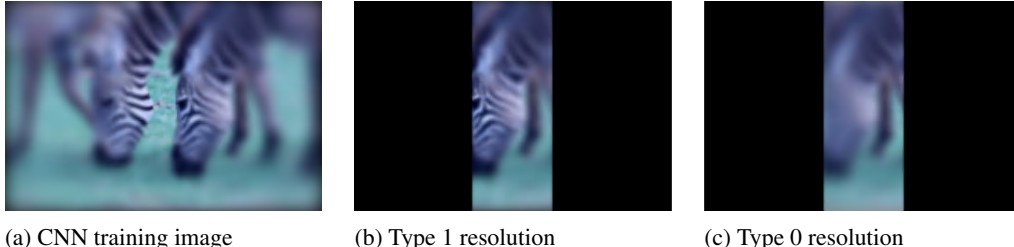

(a) CNN training image          (b) Type 1 resolution          (c) Type 0 resolution

Figure S.4: **Shifted fixation point** Example of cropped images around the center, with center fixation and shifted fixation.

Given a sample of total $5,000$ tensors (each with size $1024 \times 3 \times 1$), generated by the two different types of source images (*Type 1* and *Type 0* resolution), we want to establish whether there's a statistically significant difference between the activation pattern of the neurons *solely* depending on the resolution type. In the setting of this experiment, we have ensured that *all* other factors have been kept the same in the generation of the tensors (same object distribution between the groups etc). Formally, let $X_1, \ldots, X_{2,500}, Y_1, \ldots, Y_{2,500}$ be independent, random vectors of real numbers, representing our tensors. We have $X_i \sim \mathbb{P}_0$ and $Y_i \sim \mathbb{P}_1$, with $\mathbb{P}_0$, $\mathbb{P}_1$ probability distributions. The hypothesis test is then reduced to:

$$H_0 : \mathbb{P}_0 = \mathbb{P}_1 \quad \text{vs.} \quad H_1 : \mathbb{P}_0 \neq \mathbb{P}_1$$

Most two-sample tests assume some extent of normality. Lacking further information on the inner machinery of the training process and the distribution of kernel populations, we must establish a non-parametric, general approach where minimal assumptions are made about the data generation process. We follow Ilmun Kim (2020) and utilize the test statistic:

$$T \equiv \frac{2\hat{E}(\mathcal{C}) - 1}{\sqrt{\hat{E}_0(\mathcal{C})[1 - \hat{E}_0(\mathcal{C})]/te_0 + \hat{E}_1(\mathcal{C})[1 - \hat{E}_1(\mathcal{C})]/te_1}}$$

The hypothesis test defined by it is

$$\mathcal{H}_a \equiv \mathbb{1}(T < -z_\alpha)$$

Here $\mathcal{C} : \mathbb{X} \to \{0, 1\}$ represents a logistic-regression classifier (fit on $1,024 \times 3 \times 1 = 3,072$ variables), $te_0, te_1$ are the sizes of the test sets for each tensor type (0 or 1). We trained our classifier with 2,500 examples in total (1,250 tensors from Type 0 and Type 1). We tested it with the remaining 2,500 tensor types. *i.e.* $te_0 = 1,250, te_1 = 1,250$;

$$\hat{E}_0 \equiv \tfrac{1}{te_0} \sum_{i=1}^{te_0} \mathbb{1}(\mathcal{C}(X_i) = 1), \quad \hat{E}_1 \equiv \tfrac{1}{te_1} \sum_{i=1}^{te_1} \mathbb{1}(\mathcal{C}(Y_i) = 0), \quad \hat{E} \equiv (\hat{E}_0 + \hat{E}_1)/2$$

The test easily rejected the Null hypothesis at $\alpha = 0.01$, indicating that the two tensors were drawn from *different* distributions. We conclude that there indeed are significant differences in the activation patterns of the neurons in the network governed *solely* by resolution. This suggests that the network indeed learns separate convolutional kernels for processing objects at different resolutions, rather than a single set of uniformly applicable kernels with less precision for any given resolution type. Figure 5(d-f) shows the values of the most significant predictors learned by the Type 0/Type 1 distinguishing logistic classifier. We can clearly see that the high variable resolution images (Type 1) tend to induce a higher activation value in some neurons, while other neurons seem to fire *equally* for both resolution types. This indeed suggests some resolution specialization in the neurons of the CNN.

