# OpenReview forum: "Variable resolution: improving scene visual question answering with a limited pixel budget"
_ICLR.cc/2024/Conference — Submitted to ICLR 2024_

### Official Review · Reviewer_youK · 2023-11-01

**Soundness:** 2 fair
**Presentation:** 3 good
**Contribution:** 3 good
**Rating:** 5
**Confidence:** 4

**Summary:**

Inspired by the perspective of observing the world through biology, this article makes a bold attempt to apply a variable resolution mechanism of image on a VQA task and tries to make a trade-off between performance and computation cost. The variable resolution is implemented by a simple Log-Polar transformation while achieving a great surpass compared with the naive down-sampling strategy, and a comparable result compared with the full resolution strategy. This work then discovers the interpretability of the variable resolution, trying to prove that higher-resolution areas and lower-resolution areas can benefit from each other.

**Strengths:**

1. This paper is well written, which is reflected in its proposed variable resolution that effectively reduces the overall computational cost.
2. The innovation point of this paper is simple, and experiments on VQA and object detection have proved its effectiveness.
3. The visualization of the explanation that variable resolution performs better than uniform resolution is thorough. The attention map indicates that higher resolution areas and lower resolution areas can benefit from each other, and the neuronal activation maps and kernel filters explain the better performance from a model perspective.

**Weaknesses:**

1. Some of the experiments are missing. Firstly, as the title of the article is related to VQA, and models that perform well on object detection tasks may not necessarily have the same effect on VQA tasks, I hope to include more space in the experiment on different types of VQA datasets, such as OK-VQA. Secondly, this work has proved the effectiveness of variable resolution in that the central part has a higher resolution but only makes the comparison with uniform sampling, which is too naive to have a good performance. So I’m also curious about what the result would be when other part of the image has a higher resolution or other sampling strategies.
2. The selection strategy for high-resolution areas in the image needs to be improved. As shown in Figure 4, not all objects reside at the center part of the image, so this could be the cause of the poor performance in Table 2 when compared with the baseline. As this paper raises the HRA metric, it’s better to define a better selection strategy corresponding to this, or just sampling by gradients following the dynamic mask in [1].
[1] Lin K, Li L, Lin C C, et al. Swinbert: End-to-end transformers with sparse attention for video captioning[C]//Proceedings of the IEEE/CVF Conference on Computer Vision and Pattern Recognition. 2022: 17949-17958.
3. The interpretability analysis of variable resolution is not convincing enough. Some visualization experiments are not as important as shown in the paper, and some visualization is lacking.
On the one hand, there are too many visualization examples of the impact of the three sampling methods on the label of an object in the image. As a common sense, if we feed the model with a lower resolution of the image of the object, the model has a higher probability of giving a wrong label.
On the other hand, as shown in Figure. 3, the variable sampling scheme performs well on the questions that can be answered from parts of the image rather than the whole context of the image. That is because we don't have enough understanding of the correlation between hyperparameters and sampling strategy of variable resolution and global background knowledge. The explanation of this correlation is lacking, but the whole context of the image is important to VQA.

**Questions:**

Overall, I find the idea of this paper to be simple but reasonable. Because of its simplicity, I find no obvious weaknesses on the technical side but there is still room for improvement in performance. The authors use a large portion of this paper to explain the reason of its effectiveness, some of them are convincable, but the lack of deeper quantitative experiments makes the explanation not sufficient enough.

---

> ### Author Response · Authors · 2023-11-23
> **Additional interpretability analysis and clarifications**
>
> We thank the reviewer for the time and effort in providing this review of our paper. Following the review and very useful suggestions, we now added the evaluation of two large vision-language models on two additional datasets. We also extended the section on interpretability with a quantitative analysis of the attention spread in support of the improved performance in the VQA and detection models. We moved some details and examples from the main text to the supplementary to adhere to the 9-page limit.
> Below are answers to specific questions raised by the reviewer:
>
> **Q1.** We choose to compare between the variable scheme and the uniform-naive scheme because the uniform alternative is commonly used in many systems to make the visual input more compact and to save on compute/acquisition resources. It is, for instance, the most common arrangement of the sensors in digital cameras.
> We show that the variable scheme, which is biologically inspired, has similar compactness as the naive-uniform scheme, but allows a significant gain in model performance over the naive-uniform scheme due to the learning of efficient representations.
> In our Ablation studies (Sec.D.1), we also compared to a so-called “multi-mixed” sampling, which is a different sampling strategy.
>
> **_Action_**: We add in the revised version two new evaluations: (i) MDETR model on the GQA dataset, (ii) BLIP2 model on the SEED-Bench dataset. The results are added to Figure 3 and the supplementary. Both new evaluations support the advantage of the variable scheme over the uniform scheme, as already demonstrated in the evaluation on the VQAv2 dataset.
>
>
> **Q2.** Indeed, as Table 2 shows, the baseline outperforms the variable scheme, in part due to the fact that many objects are not included in the high-resolution area. However, this paper focuses on the representational differences learned by different sampling schemes. To make our analysis more general and agnostic to specific scene layouts, we intentionally choose an arbitrary fixation point, which is not targeting a specific object or salient feature in the scene. As can be seen in Figure 4, the variable scheme gain in performance over the uniform scheme even when objects are mostly in the periphery (up to 70% of an object’s area) and only a small part is included in the high-resolution area (image center in our settings). Figure 5 presents the representational roots of the model performance gain discussed in section 4. Certainly, a good fixation strategy can improve performance, however in this paper we aim to establish solid single-fixation foundations for future works exploring this venue in complex VQA.
>
>
> **Q3.** **_Action_**: In the revised paper, we provide an additional quantitative analysis in the interpretability section, in particular the attention distance measure for the self-attention layers of the DETR model (also used for VQA). This analysis indicates that indeed the variable scheme utilizes contextual information from the far periphery (in particular, in intermediate layers 2 and 3), while the uniform scheme only utilizes information from adjacent spatial areas.
> We also add evaluations on the VQA task in Figure 3, which clearly shows a consistent advantage of the variable scheme over the uniform scheme in questions requiring the integration of fine details and contextual information, such as in questions about object relations and object classification (including object attributes).

---

### Official Review · Reviewer_hgaY · 2023-11-04

**Soundness:** 2 fair
**Presentation:** 3 good
**Contribution:** 2 fair
**Rating:** 5
**Confidence:** 5

**Summary:**

The authors claim that they propose a variable sampling scheme, inspired by human vision, remarkably outperforms a uniform sampling scheme by 2% accuracy (65% vs. 63%) in the challenging task of scene visual question answering (VQA), under a limited samples budget (3% of the full resolution baseline).

**Strengths:**

The authors claim that they propose a variable sampling scheme, inspired by human vision, remarkably outperforms a uniform sampling scheme by 2% accuracy (65% vs. 63%) in the challenging task of scene visual question answering (VQA), under a limited samples budget (3% of the full resolution baseline).

**Weaknesses:**

1. According to the title, the author's focus is on VQA. but it seems that only one dataset (VQA v2) from VQA task was used. How does the authors' proposed method perform on other common datasets such as GQA[1], OOD-GQA[2], etc.?

[1] Hudson, Drew A., and Christopher D. Manning. "Gqa: A new dataset for real-world visual reasoning and compositional question answering." Proceedings of the IEEE/CVF conference on computer vision and pattern recognition. 2019.

[2] Kervadec, Corentin, et al. "Roses are red, violets are blue... but should vqa expect them to?." Proceedings of the IEEE/CVF Conference on Computer Vision and Pattern Recognition. 2021.


2. The method proposed by the authors seems to address object detection and is not a VQA task.

3. Does Sec.5 relate to the VQA task? Why?

4. Is the technical or scientific challenge addressed throughout the article associated with VQA? Why?

**Questions:**

Please refer to Weakness.

---

> ### Author Response · Authors · 2023-11-23
> **Experiments and clarifications added**
>
> We thank the reviewer for the time and effort in providing this review of our paper. We found the remarks very useful and now added the evaluation of two large vision-language models on two additional datasets. We also extended the section on interpretability with a quantitative analysis of the attention spread in support of the improved performance in the VQA and detection models.
> We moved some details and examples from the main text to the supplementary to adhere to the 9-page limit.
> Below are answers to specific questions raised by the reviewer:
>
> **Q1.** **_Action_**: We added in the revised version two new evaluations of the model on challenging VQA tasks. In particular, we included an evaluation for the MDETR vision-language model (VLM) on the challenging GQA dataset. In addition, we included a third evaluation of the BLIP2 VLM on the new SEED-Bench dataset. We added a summary graph of the results to Figure 3. Both new evaluations show a similar advantage of the variable scheme over the uniform scheme, as already demonstrated in the evaluation on the VQA dataset, supporting the claim that this is a consistent advantage of variable resolution in VQA tasks.
>
>
> **Q2.** It may be too challenging to analyze the contribution of the mechanisms combined in a complex VLM, in particular the visual mechanism. Therefore, we choose to investigate underlying visual sub-tasks that drive the improvement in performance for the VLMs when applied to VQA tasks. As mentioned above (Q1), we included in the revised version, an evaluation of the MDETR VLM, which is based on the architecture of the DETR detection model combined with a BERT-based LLM. In sections 4 and 5 we analyze in detail the performance and interpretability of the DETR model, which provide direct insights on the visual representations of the MDETR.
>
>
> **Q3.** Yes, following the reviewer's relevant remarks, Sec.5 is now directly related to our findings on VQA. To clarify why, the reason is that the entire analysis on the attention was done on the DETR model, which we apply to the GQA dataset (with MDETR) and achieve significant results, consistent with our findings on VQAv2. The analysis of the CNN kernel specialization was done on ResNet101, which is the backbone of MDETR and DETR.
>
>
> **Q4.** The VQA task is closely related to the task of complex visual scene understanding, for which we believe the contribution of a variable scheme is potentially highly significant. Understanding object relations and interactions in the scene, in addition to particular object attributes, requires the integration of fine visual cues with contextual information from the scene. Both are available with the variable resolution scheme. We find the evaluation results of VLMs on the task of VQA, when applied to the variable sampling scheme, significant as they clearly show that these models mostly benefit from the variable sampling scheme over the uniform scheme. However, as mentioned above (Q2), it is not clear how to analyze the contribution of the visual mechanism within a complex VLM, which combines a large language mechanism. We therefore investigate representational learning in basic visual sub-tasks, including image classification and object detection, which underlie and drive the visual capacities of a VLM.

---

> > ### Comment · Reviewer_hgaY · 2023-11-23
> >
> > The author has answered my question, but I don't find the relevant sentence in the revised version. Therefore I keep my rating.

---

### Official Review · Reviewer_V3rN · 2023-11-06

**Soundness:** 3 good
**Presentation:** 3 good
**Contribution:** 3 good
**Rating:** 6
**Confidence:** 5

**Summary:**

The authors re-evaluate the use an design of a variable resolution visual sensor for neural networks, and try to approach this problem from the angle of representational gains rather than computational efficiency (what is classically accepted in the literature). Authors show several experiments ranging from detection, VQA + interpretability to show the advantage when 2 systems are placed under equal perceptual sensing conditions to perform inference (uniform vs variable) and thus finding that the neural network with a variable resolution sensor in many cases out-performs the equal resolution one (that is usually more blurred).

**Strengths:**

* The paper tackled the question of the use of a foveated (spatially-adaptive) visual system through the lens of object detection, interpretability + Visual Question Answering (VQA). I will give it to the authors, as I don't think this has ever been done before, which is why I am marginally inclined to accept this paper. Most methods of testing for the representational goal of foveation is through object classification or detection, and even more recently for texture-based discrimination. Authors talk a bit about this too, in addition to present interpretabilty experiments similar to Deza & Konkle. ArXiv, 2021.
* Authors have made a good case for showing representational gains of a foveated visual system

**Weaknesses:**

There is a long list of critical missing papers that should be cited if this paper is to be accepted. I can not increase my score to accept unless these papers are cited & discussed (and of course, if the other reviewers also think that this paper should be accepted).

Key Missing Critical References:
- Deza & Konkle. ArXiv, 2021. **Emergent Properties of Foveated Perceptual Systems.**
- Wang & Cottrell. Journal of Vision, 2017. **Central and peripheral vision for scene recognition: A neurocomputational modeling exploration.**

Secondary, but also important References:
- Cheung, Weiss & Olshausen. ICLR 2017. Emergence of foveal image sampling from learning to attend in visual scenes
- Gant, Banburski & Deza. SVRHM, 2022. Evaluating the adversarial robustness of a foveated texture transform module in a CNN.
- Reddy, Banburski, Pant & Poggio. NeurIPS 2020. Biologically inspired mechanisms for adversarial robustness
- Wang, Mayo, Deza, Barbu & Conwell. SVRHM, 2021. On the use of Cortical Magnification and Saccades as Biological Proxies for Data Augmentation
- Harrington & Deza. ICLR, 2022. Finding Biological Plausibility for Adversarially Robust Features via Metameric Tasks
- Malkin, Deza & Poggio. SVRHM 2020. CUDA-Optimized real-time rendering of a Foveated Visual System.

----

**Questions:**

* While I think the VQA evaluation framework is original, what I do not understand is "why VQA?", why not something less complex such as object recognition or detection where language modelling will not interfere in the output produced by the system. I can only think of an answer if there is an argument somehow linking foveation with language but that does not seem to be the case. I can see an object detection evaluation which I think is nice, but that has already been shown in Pramod et al. 2021.

* Comparison of this work with Deza & Konkle is necessary. They addressed many questions presented in this paper such as training on different types of foveal-peripheral transforms, the use of foveation as a texture-based distortion that mimics crowding vs a more rudimentary baselines such as adaptive gaussian blurring. Furthermore it would have been interesting if the Authors would have compared their results in ImageNet vs Places. Does the same pattern of results hold? Does Foveation do better by virtue of a central image bias where the object is usually put in the center of the image? Presumable more controlled experiments are necessary to try to answer these questions.

* I am not sure what the red dots represent in many of the figures such as Figure 5 and S5.

All in all, I think this paper is exciting, but discussing and adding the missing references is necessary.

---

> ### Author Response · Authors · 2023-11-23
> **References and discussions added**
>
> We thank the reviewer for the time and effort in providing a useful review of our paper.
> Following the reviews we now added and discussed the missing references in the Related work section. Additionally, we now include the evaluation of two large VLMs on two additional datasets. We also extended the section on interpretability with a quantitative analysis of the attention spread in support of the improved performance in the VQA and detection models. We moved some details and examples from the main text to the supplementary to adhere to the 9-page limit.
> Below are answers to specific questions raised by the reviewer:
>
> **Q1.** The task of visual question answering is closely related to the task of complex visual scene understanding, for which we believe the contribution of a variable resolution system is potentially more significant than for detection alone.
> The basic tasks including object classification and detection are important and indeed are more convenient to investigate with respect to the representational learning patterns, as we do in Sections 4 and 5. Nevertheless, the integration of fine cues at high resolution with contextual information at lower resolutions may be limited compared to the task of VQA, since the basic tasks do not often require the understanding of subtle attributes, relations and interactions in the image. This level of integration between fine and global information is available with the variable resolution scheme.
> We find the evaluation results of VLMs on the task of VQA significant as they clearly show that these models mostly benefit from the variable sampling scheme over the uniform scheme.
>
> **_Action_**: We added a similar clarification in the revised paper.
>
>
> **Q2.1.** We thank the reviewer for pointing out a number of important papers. Indeed, we did not cover many studies around the topic of foveated images, as we try to focus on the contribution of variable resolution to high-level complex tasks, including visual scene understanding (question answering) and object recognition (detection), already at the initial fixation point in the scene. To make our analysis more general and agnostic to specific scene layout, we intentionally choose an arbitrary fixation point, which is not targeting a specific object or salient feature in the scene. We provide a very extensive analysis of this initial fixation performance and hope to establish a solid foundation for future works exploring VQA with re-fixations.
>
> **_Action_**:  We added in the revised version a discussion to section 6 on the important papers we were referred to by the reviewer.
>
>
> **Q2.2.** Regarding the inquiry concerning the central image bias, our bin experiment, presented in Figure 4 and described in Section 4, shows that foveation does better than uniform even when more than half of the object is placed in the periphery, while only a small portion of the object lies at the center of fixation in the image. As an additional control, we also tested the detection models on a validation set consisting of objects encompassed by the exact same number of visual samples (see Table 2). The DETR model, tested on the above two controls, was actually used in the VQA as part of the MDETR model.
>
>
> **Q3.** Figure 5 and S5 include example visualizations of the affinity (or “interest”) that the transformer query tokens exhibit towards different parts of the image. This is analogous to the activation of a receptive field in a CNN - the higher the intensity on the attention map, the more “attentive” the token is to that part of the image.
> The red dots overlaid on the images show the location of the particular tokens we chose (in the periphery) and the corresponding attention map around the image is depicted closest to the red dot.
>
> **_Action_**: We provided a clarification in the figure captions of the revised version.

---

### Meta-Review · Area_Chair_6BpP · 2023-12-14

**Metareview:**

This paper explores the use of a variable resolution visual sensor in neural networks for scene visual question answering (VQA). It aims to balance computational efficiency with representational gains by employing a foveated, or spatially-adaptive, visual system. Reviewer V3rN acknowledged the novelty and representational gains of the proposed system but suggested a few missing references. They provided a marginally positive score. Reviewer hgaY raised concerns about the paper's focus and the absence of evaluations on diverse datasets like GQA and OOD-GQA. The authors somewhat addressed the concerns. Reviewer youK appreciated the paper's simplicity and empirical results but pointed out the lack of deeper quantitative experiments and analyses.

In summary, the paper lacks a strong champion support and the collective feedback from reviewers tends towards a marginal rejection. After thoroughly reviewing the paper, the Area Chair agrees with this view. For such a simple idea, it is essential to enhance both the clarity of its presentation and the thoroughness of its analysis. Currently, it's challenging to assess the possible impact this research might have in the field of multimodal learning. While the paper definitely has merits, especially in its novel approach and additional improvements made during the review process, it may not have fully met the standards expected for acceptance and requires a major revision.

**Justification For Why Not Higher Score:**

The consensus among reviewers and the Area Chair leans towards a rejection recommendation. Significant revisions are needed to enhance both the quality/clarity of the presentation and the depth of the experiments and analyses.

**Justification For Why Not Lower Score:**

N/A

---

### Decision · Program_Chairs · 2024-01-16

Reject